# LPNet: Retina Inspired Neural Network for Object Detection and Recognition

**Jie Cao** [1,2], **Chun Bao** [1,2], **Qun Hao** [1,2,*], **Yang Cheng** [1,2,*] and **Chenglin Chen** [1,2]

[1] Key Laboratory of Biomimetic Robots and Systems, Ministry of Education, Beijing Institute of Technology, Beijing 100081, China; caojie@bit.edu.cn (J.C.); baochun@bit.edu.cn (C.B.); 3120190616@bit.edu.cn (C.C.)
[2] Yangtze Delta Region Academy, Beijing Institute of Technology, Jiaxing 314003, China
* Correspondence: qhao@bit.edu.cn (Q.H.); yangcheng2007@163.com (Y.C.)

**Abstract:** The detection of rotated objects is a meaningful and challenging research work. Although the state-of-the-art deep learning models have feature invariance, especially convolutional neural networks (CNNs), their architectures did not specifically design for rotation invariance. They only slightly compensate for this feature through pooling layers. In this study, we propose a novel network, named LPNet, to solve the problem of object rotation. LPNet improves the detection accuracy by combining retina-like log-polar transformation. Furthermore, LPNet is a plug-and-play architecture for object detection and recognition. It consists of two parts, which we name as encoder and decoder. An encoder extracts images which feature in log-polar coordinates while a decoder eliminates image noise in cartesian coordinates. Moreover, according to the movement of center points, LPNet has stable and sliding modes. LPNet takes the single-shot multibox detector (SSD) network as the baseline network and the visual geometry group (VGG16) as the feature extraction backbone network. The experiment results show that, compared with conventional SSD networks, the mean average precision (mAP) of LPNet increased by 3.4% for regular objects and by 17.6% for rotated objects.

**Keywords:** convolutional neural networks; LPNet; retina-like; log-polar; object detection and recognition

## 1. Introduction

In recent years, deep learning has played an important role in many areas of life, such as image processing [1–3], object detection [4–6], optic imaging [7–9], and speech recognition [10,11]. Especially in object detection and recognition, the accuracy of deep learning models becomes increasingly important [12]. However, due to the dependence on datasets, deep learning networks have limitations for special objects, such as rotated objects. The rotation invariant feature is one of the key methods to solve the problem of targeting variable direction. For example, remote sensing images cause difficulties for conventional convolutional neural network models to achieve robust detection.

For rotated objects, features mismatching is a challenging work [13], which will reduce the final recognition accuracy. There is a large number of deep learning methods to solve such problems [14,15]. In fact, some works solved these problems by the retina-like mechanism [16–18]. In the human visual system, photoreceptor cells of the retina are sparsely distributed as the distance from the fovea increases. This structural characteristic causes the high imaging resolution of the retina in the vicinity of the fovea and low imaging resolution in the peripheral area [19]. Between the retina and the visual cortex, a log-polar coordinate mapping relationship is established [20] and brings the advantages of anti-rotation and scale transformation [21–24].

Based on these characteristics of human retinal imaging, the retina-like mechanism is used in various fields [25–27]. For example, in the task of object search and detection in a large field of view and high-resolution scene, the data are compressed using log-polar

coordinates with variable resolution imaging. A retina-like mechanism is also widely used in high-speed rotation navigation and guidance [28]. Similarly, numerous studies use the retina-like log-polar transformation in the classification task of deep learning [29–31]. However, for object detection and recognition, few works combine log-polar transformation with convolutional neural networks (CNNs) [32]. Based on these methods, we propose a log-polar coordinates feature extraction network, named LPNet. Inspired by the principle of human retinal imaging, we introduce the log-polar transformation that overcomes the rotation problem into CNN. We also put the variable resolution characteristics of the human eye into LPNet, which effectively increases the robustness of networks.

Several works address the space invariance on image processing using CNNs [33–35]. In [36], log-polar transformation is brought into the field of deep learning through the polar transformation network (PTN), which realizes the invariance of translation, the equal change in rotation, and the expansion in the polar coordinate system. However, PTN only recognized global deformation. In [37], the results show that several angles are better fitted to CNNs with log-polar operations for all tested datasets. However, only rotation transformation is performed and experiments are carried out under different rotations. In [38], log-polar is combined with the attention mechanism. Thus, several improvements are achieved from the original basis. A large amount of experimental verification is lacking and they just focused on classification tasks. In [39], they applied a log-polar transformation as a pre-processing step to a classification CNN. It reduced the required image size and improved the performance in handling image rotation and scaling permutations. However, this method is applied using the MNIST dataset for classification. Object detection and recognition on other datasets are not discussed. The above research works show that log-polar gradually established its role in deep learning, but the transformations are rarely applied to object detection and recognition networks.

The rest of this paper is organized as follows. Section 2 introduces the preliminary background knowledge. Section 3 describes the proposed method in detail. Section 4 presents the experimental settings. Finally, Section 5 reports the general conclusions and suggests future research directions.

## 2. Background

### 2.1. Log-Polar Transformation

For an image, if we select the coordinate origin as O (0, 0), the position of the pixel based on sampling is expressed in both cartesian coordinates $(x, y)$ and log-polar coordinates $(r, \theta)$ [40–43]. Equations (1) and (2) show the conversion between two coordinates.

$$r = \sqrt{x^2 + y^2} \tag{1}$$

$$\theta = \arctan(y/x) \tag{2}$$

In the cartesian coordinate system, $z$ is a complex number. We define the equation for the log-polar coordinate, as shown in Equation (3).

$$w = \ln z \tag{3}$$

We set a cartesian coordinate number represented by the value of $z = x + iy$. Log-polar coordinates are represented by the value of $w = \xi + i\eta$, where $i$ is the complex imaginary unit.

$$z = x + iy = r(\cos\theta + i\sin\theta) = re^{i\theta} \tag{4}$$

$$w = \xi + i\eta = \ln z = \ln r + i(\theta + 2\pi) \tag{5}$$

Log-polar coordinates $\xi$ and $\eta$ are given as Equations (6) and (7).

$$\xi(r, \theta) = \ln r = \ln\sqrt{x^2 + y^2} \tag{6}$$

$$\eta(r,\theta) = \theta + 2\pi = \arctan(y/x) + 2\pi \tag{7}$$

The points with the coordinate $(x, y)$ are mapped to the coordinate $(\xi, \eta)$ after transformation.

According to the above equations, the object in log-polar coordinates $w$ changes in proportion and rotation. That means that the object is magnified by $\tau$ times to the origin of the coordinate. If the objects are rotated by an angle of $\alpha$, then its corresponding polar coordinate changes to $(\tau r, \theta + \alpha)$. After log-polar transformation, the mapping is as Equations (8)–(10).

$$w = \ln z = \ln \tau r e^{i(\theta + \alpha)} \tag{8}$$

$$\xi(r,\theta) = \ln r + \ln \tau \tag{9}$$

$$\eta(r,\theta) = \theta + \alpha + 2k\pi \tag{10}$$

If the target in the cartesian coordinates changes in proportion, then it is equivalent to the target in the log-polar coordinates being displaced along the radius axis [37]. The rotation change in the target in cartesian coordinates is equivalent to the displacement change in the target in the log-polar coordinate space along the angular axis. We take $2\pi$ as the periodic displacement, the log-polar transformation has scale and rotation invariance [44] which is established when the center point coincides with the coordinate origin after the target change [45].

According to these characteristics, we introduce log-polar into object detection and recognition network to alleviate the influence of rotating targets on accuracy. During image rotation or transformation, the same object will have different feature maps after CNN extraction. Therefore, the image becomes inconsistent with the results of conventional network training. To solve the problem of reduced accuracy due to image rotation, we introduce log-polar transformation. Moreover, we call the log-polar transformation for feature maps during the encoding process. After performing a convolution operation, an inverse log-polar transformation is carried out on the network, which is called the decoding process of CNN.

As shown in Figure 1a–d, when the target in the images is rotated by 90°, the feature of the encoded image remains unchanged. Compared to Figure 1b,c, the feature map encoding shows that the rotation only performs translation transformation. However, the distribution of feature maps remains unchanged. For CNNs, this slight change does not cause significant accuracy falling.

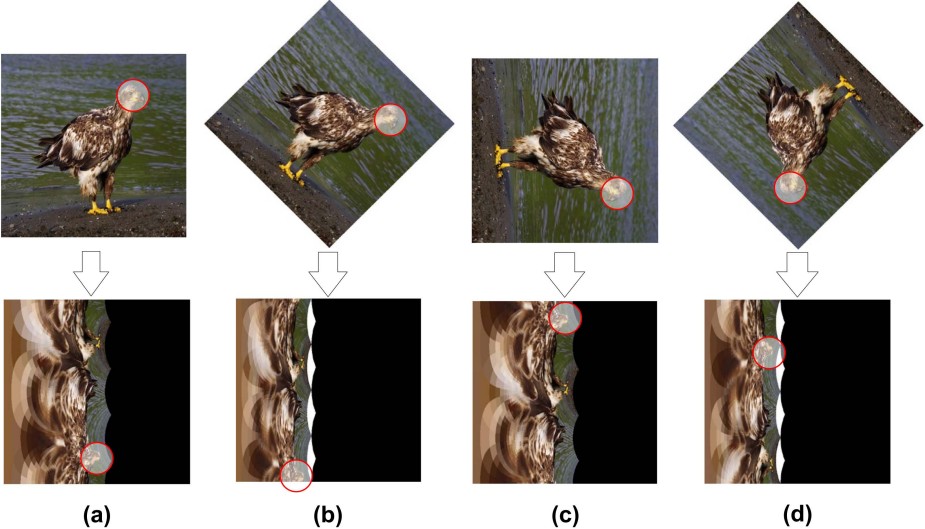

**Figure 1.** Log-polar transformation for a rotated target: (**a**) Original image and encoded image; (**b**) original image rotated by 45° and the encoded image; (**c**) original image rotated by 90° and encoded image; (**d**) original image rotated by 135° and encoded image.

*2.2. SSD300*

The baseline network of the proposed LPNet is the single-shot multibox detector (SSD). SSD is an objection detection network proposed by Liu et al. [46] and is one of the popular frameworks. Figure 2 shows the architecture of SSD300. Compared with the faster region-based convolutional network (R-CNN) [47], the SSD network has a considerable speed improvement. Compared with you only look once (YOLO) network [48], SSD has a clear advantage in accuracy. SSD uses VGG16 as the backbone network.

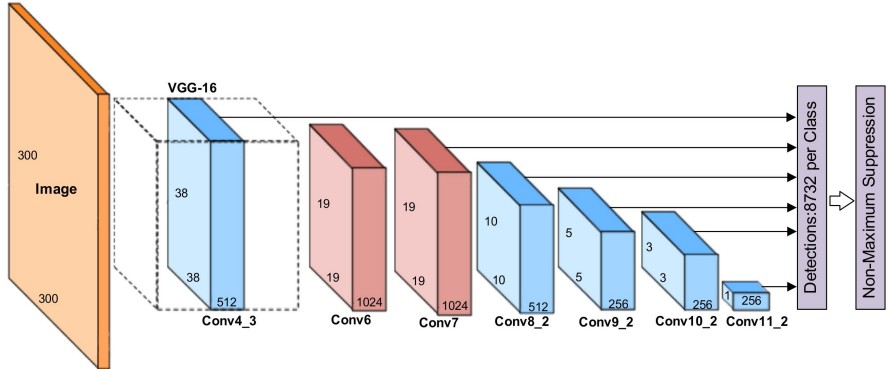

**Figure 2.** The architecture of SSD300. We select it as the baseline network.

The obtained multi-scale feature maps are used for the detection of objects. In the process of locating objects, large feature maps are used on relatively small objects while small feature maps are used for large ones. SSD refers to YOLO but differs from it. It uses convolution to detect feature maps of different scales and, at the same time, draws on the idea of anchors in faster R-CNN.

The speed advantage of SSD lies in the fact that the algorithm is implemented based on the feedforward architecture. The calculations are all in an end-to-end single channel. For a single input image, SSD generates multiple fixed-size bounding boxes and scores for the object category. Then, non-maximum suppression (NMS) operation is added to obtain the final prediction. Thus, the detection speed is significantly improved. The first half is the basic network, which is mainly used for image classification. The second half is a multi-scale convolutional layer with a size reduced layer by layer and is mainly used for the extraction and detection of object features at multiple scales.

## 3. LPNet

*3.1. Architecture Overview*

In this section, we extract feature maps from the backbone network and transform them to log-polar coordinates. The baseline framework is the SSD300 network and the feature extraction backbone network is VGG16. Figure 3 shows the structure of the proposed LP layer.

We regard the module that combines convolution and log-polar in the feature extraction as the encoder. However, the module that combines the convolution unit and inverse log-polar is the decoder. The ratio of numbers of feature maps that require a log-polar transformation in each layer is defined as the observation factor (*OF*), which affects the overall network performance. We will discuss this in Section 4.

In the decoder module, the usual cartesian coordinates are converted to log-polar coordinates. This transformation helps the network extract the feature information in the log-polar coordinates of the feature map. When the object is rotated and scaled, the network error effectively decreases. Equations (11) and (12) show the calculations of the conventional CNN and the encoder module, respectively. However, this transformation also raises several problems. After the encoder module, all information extracted by the CNN is of log-polar coordinates. The feature information of the image itself cannot be extracted further. To solve this problem, we introduce the inverse log-polar module.

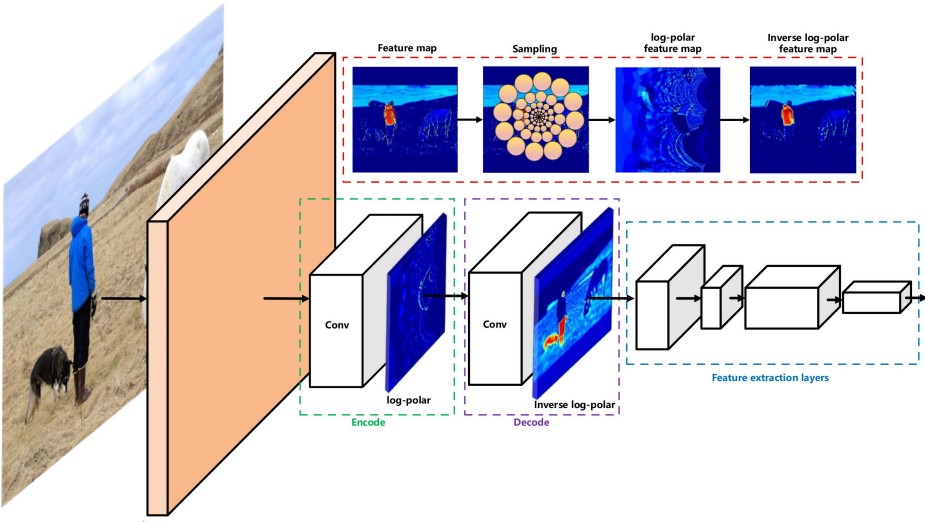

**Figure 3.** The architecture of the LP layer. In the encoder module, the input feature maps are normal uniform pixel distribution images. After the log-polar transform, the feature maps are resampled as nonuniform distribution. Through inverse log-polar transform, we get the "inverse log-polar feature map". The pixels in this map are sampled as human eyes. This image has high resolution in the center and low resolution on the side.

$$g(x_o, y_o) = \int_{-\infty}^{+\infty} \int_{-\infty}^{+\infty} f(x,y) h(x_o - x, y_o - y) dx dy \qquad (11)$$

$$g(r_o, \theta_o) = \int_0^{2\pi} \int_0^\infty f(r,\theta) h(r_o \cos\theta_o - r\cos\theta, r_o \sin\theta_o - r\sin\theta) r dr d\theta \qquad (12)$$

where $g(x_o, y_o)$ is the output feature map in cartesian coordinates, $g(r_o, \theta_o)$ is the output feature map in log-polar coordinates, $f(\cdot)$ is the input feature map, and $h(\cdot)$ is the convolution kernel.

In this module, we convert the cartesian coordinates of the feature map to log-polar coordinates. The feature maps in the log-polar coordinates have more characteristics than the cartesian coordinates, such as rotation invariance, and so on. We show this transformation in Figure 4.

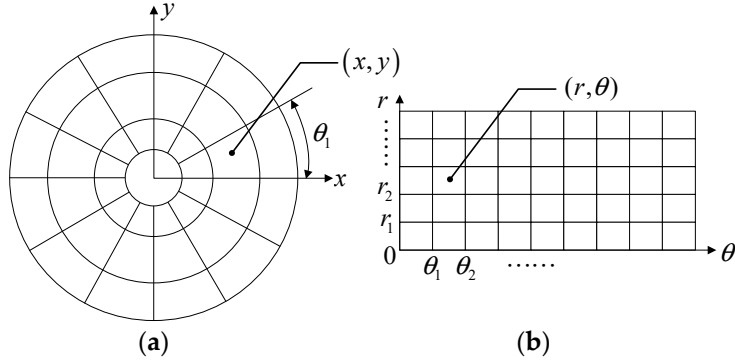

**Figure 4.** The mapping from cartesian coordinates to log-polar coordinates. We convert the cartesian coordinates (**a**) of the feature map to log-polar coordinates (**b**).

In the decoder module, we add an inverse transformation of the feature map after the log-polar transformation. The log-polar coordinate information is converted into the image information in cartesian coordinates to facilitate the subsequent feature extraction of the CNN.

This operation is the opposite of the transformation in Figure 4. We rearrange the pixels in log-polar coordinates on a new feature map. While the feature points of log-polar coordinates and the feature points under the cartesian coordinates are not in a one-to-one correspondence, we dropped some points on the original feature map. After the decoding, the points of interest in the feature map show high resolution while other regions appear in low resolution. This result is also similar to the observation characteristic of the human eye, which is non-uniform imaging. This characteristic also alleviates the influence of noise points during the convolution operation. Thus, the interference of redundant features on the network accuracy is suppressed.

### 3.2. Sliding LPNet

With the consideration that feature extraction by the human eye scans to the extracting method, the starting point of the high-resolution field is the center of the image. However, the interesting area is not all in the center. Furthermore, the objects scatter everywhere in an image. Therefore, in this section, we propose another scanning feature map gaze center selection method.

We divide the feature map into small units according to the size ratio. Before the log-polar transformation is performed on each feature map in each layer, the center of each small cell is used as the center of the log-polar transformation of entire feature maps. The position of the center point coordinate relative to entire feature maps is sequentially presented in a sliding window format. Figure 5 shows the splitting of small cells in the feature map, the size of which determines the center coordinate of each small cell. The selected center point is used as the basis for the next feature map encoding.

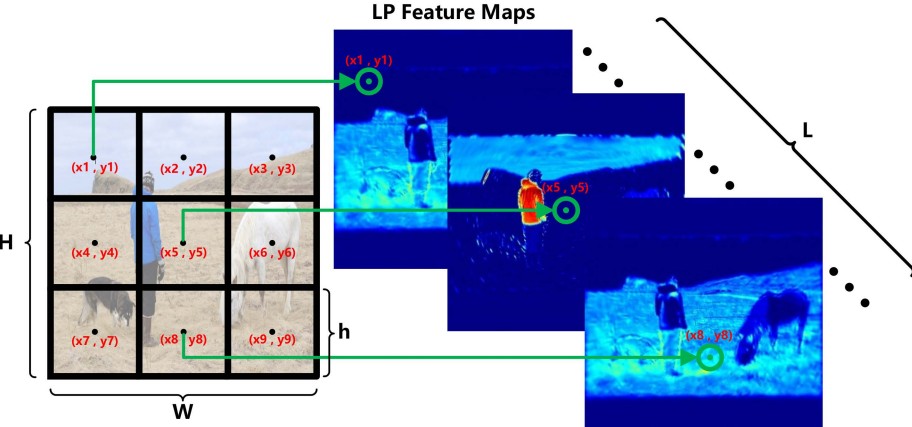

**Figure 5.** LPNet in the mode of sliding centers. In different channels, we choose different points as the center of log-polar transformation. That means the high-resolution area in images is sliding.

The number of grids to be cut is $\lambda \times \lambda$. Then, the image is cut into small grids $h \times h$. The size of the original image is $H \times W$. In Figure 5, we take $H = W = 300$, $h = 100$ as an example, and take the center point of each small grid as the center point of each non-uniform sampling. According to the setting of *OF* value, the number of feature maps that need to be coded and decoded is calculated as $L$. A center point selection on the $L$ feature maps is carried out in turn and calculated according to the coordinate of the cutting. Table 1 shows the obtained coordinates.

**Table 1.** Center coordinates of LPNet, where $H = W = 300$.

| Location | Value | Location | Value | Location | Value |
|---|---|---|---|---|---|
| (x1,y1) | (50,50) | (x4,y4) | (50,150) | (x7,y7) | (50,250) |
| (x2,y2) | (150,50) | (x5,y5) | (150,150) | (x8,y8) | (150,250) |
| (x3,y3) | (250,50) | (x6,y6) | (250,150) | (x9,y9) | (250,250) |

## 4. Experimental Results and Analysis

In this section, we take the SSD network as the baseline framework. According to the experimental results, the increase in accuracy is also suitable for other networks.

### 4.1. Experiment Setup

In object detection and recognition experiments, the default hyper-parameters are as follows: the training step is 120,000; the training and testing batch sizes are 32 and 64, respectively; the polynomial decay learning rate scheduling strategy is adopted with an initial learning rate of 0.1; the warm-up step is 1000; and the momentum and weight decay are 0.9 and 0.005, respectively. All of our LPNet experiments use the same hyper-parameter as the default setting. All experiments are trained with two 2080Ti GPUs.

We trained LPNet on the pattern analysis, statical modeling, and computational learning (PASCAL) VOC 2007 and PASCAL VOC 2012 datasets. The main goal of the PASCAL VOC dataset is to recognize image objects. The dataset contains 20 types of objects. Then, each image is labeled. The labeled objects include people, animals (such as cats, dogs, islands, etc.), vehicles (such as cars, boats, airplanes, etc.), and furniture (such as chairs, tables, sofas, etc.). Each image has 2~4 objects on average. All the annotated images have labels needed for object detection and recognition, but only some data have segmentation labels. Among them, the VOC2007 dataset contains 9963 annotated images. These images are divided into three categories, consisting of train, val, and test. Then, the total 24,640 objects are labeled. The VOC2012 dataset is an upgraded version of the VOC2007 dataset, with a total of 11,530 images. For the objection detection, the trainval/test of VOC2012 contains all the corresponding images from 2008 to 2011 years. The trainval has 11,540 images and a total of 27,450 objects. For the segmentation task, the trainval of VOC2012 contains all the corresponding images from 2007 to 2011 years, and the test only contains 2008 to 2011 years. Trainval has 2913 images with 6929 objects. The VOC2012 dataset is divided into 20 categories, including 21 categories of background.

### 4.2. Results

We carried out experiments on the PASCAL VOC2007 and PASCAL VOC2012 datasets, and divided the LPT mode into two types (sliding and stable). During the network training, VOC2007 + VOC2012 is used as the training set and tested on the VOC2007 test dataset. Table 2 shows the test results.

**Table 2.** Experiment results of LPNet in terms of mAP (%) on PASCAL VOC 07 + 02. Where "07 + 12" means the union of VOC2007 and VOC2012 trainval.

| Method | mAP | Aero | Bike | Bird | Boat | Bottle | Bus | Car | Cat | Chair | Cow | Table | Dog | Horse | mBike | Person | Plant | Sheep | Sofa | Train | Tv |
|--------|-----|------|------|------|------|--------|-----|-----|-----|-------|-----|-------|-----|-------|-------|--------|-------|-------|------|-------|-----|
| LPNet(sliding) | 77.6 | 81.8 | 84.5 | 76.2 | 71.9 | 49.4 | 85.9 | 86.3 | 87.5 | 62.0 | 81.6 | 73.9 | 86.1 | 85.8 | 84.1 | 79.8 | 53.7 | 78.4 | 80.8 | 85.7 | 76.7 |
| LPNet(stable) | 77.7 | 82.8 | 83.8 | 75.7 | 71.0 | 51.1 | 86.0 | 86.1 | 88.3 | 62.7 | 81.5 | 84.2 | 86.2 | 87.3 | 84.2 | 79.5 | 53.3 | 77.7 | 80.7 | 85.3 | 77.2 |
| SSD300 [46] | 74.3 | 75.5 | 80.2 | 72.3 | 66.3 | 47.6 | 83.3 | 84.2 | 86.1 | 54.7 | 78.3 | 73.9 | 84.5 | 85.3 | 82.6 | 76.2 | 48.6 | 73.9 | 76.0 | 83.4 | 74.0 |
| Faster [47] | 73.2 | 76.5 | 79.0 | 70.9 | 65.5 | 52.1 | 83.1 | 84.7 | 86.4 | 52.0 | 81.9 | 65.7 | 84.8 | 84.6 | 77.5 | 76.7 | 38.8 | 73.6 | 73.9 | 83.0 | 72.6 |
| Fast [49] | 70.0 | 77.0 | 78.1 | 69.3 | 59.4 | 38.3 | 81.6 | 78.6 | 86.7 | 42.8 | 78.8 | 68.9 | 84.7 | 82.0 | 76.6 | 69.9 | 31.8 | 70.1 | 74.8 | 80.4 | 70.4 |

In this experiment, the performance indicators are mainly compared with SSD and similar networks. Compared with the high-performance network that appeared after SSD, experimental results also draw the same inference. Table 2 shows that, under VOC2007 and VOC2012 datasets, two modes of LPNet have higher accuracy than SSD300, faster R-CNN, and fast region-based convolutional network (R-CNN) [49].

Compared with the fast R-CNN algorithm, the mAP of LPNet is improved by 7.6%. This improvement comes from the fact that LPNet can effectively reduce the influence of noise on the input feature map. Moreover, for objects of different sizes, the detection and recognition accuracy of LPNet improved. Consistent with the input image size of SSD300, both modes of LPNet choose 300×300 as the image input size of the network. In the network training, Figure 6 shows the loss and mAP of the modes. Figure 7 shows that

the LPNet of the sliding mode is more steady than that of the stable mode, and that the mAP is improved.

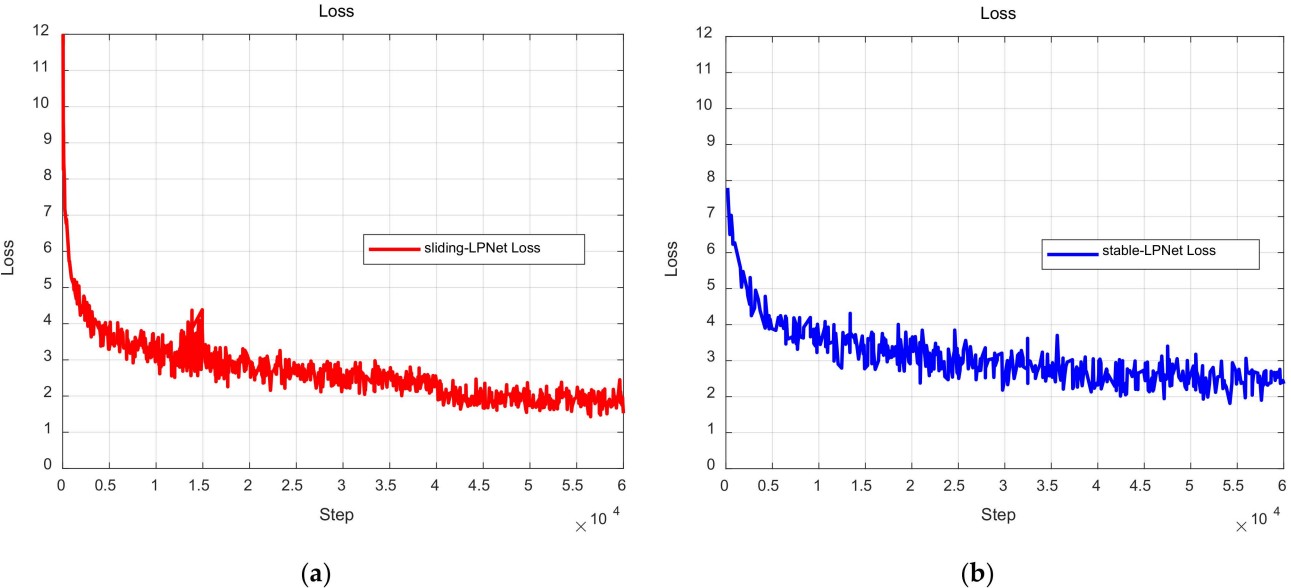

(**a**)  (**b**)

**Figure 6.** Two modes of LPNet Loss, where the two networks are trained under the VOC2007 and VOC2012 datasets, and the *OF* = 1: (**a**) Loss of LPNet in sliding mode; (**b**) Loss of LPNet in stable mode.

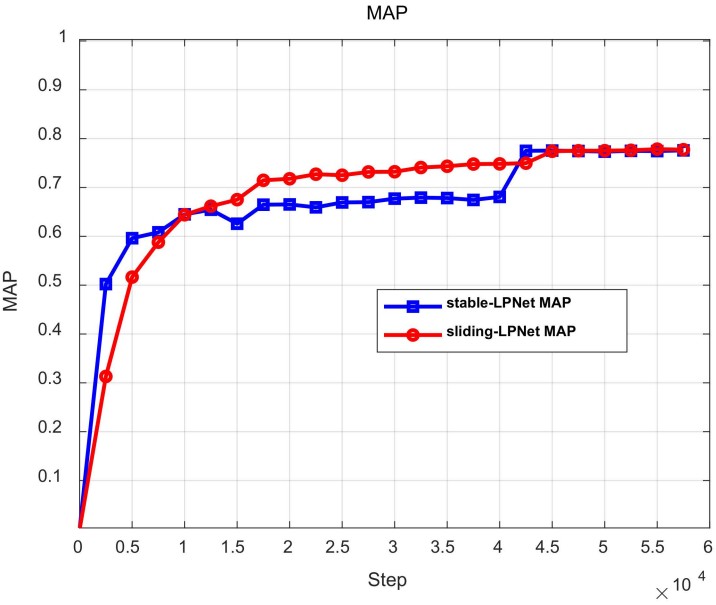

**Figure 7.** mAP of two modes of LPNet during the training, with the final accuracy of the sliding mode LPNet as slightly higher than that of the stable mode LPNet.

### 4.3. OF Impact

The *OF* value represents the intensity information of the retina-like mechanism on CNN. This parameter also determines the number of feature maps that need focus. To study the influence of the *OF* on the overall network performance of the network, we select its gradient values. For two different modes of LPNet, the mAP test is performed on the VOC2007 + VOC2012 dataset. Table 3 shows the final model accuracy.

**Table 3.** Accuracy of LPNet in gradient observation factors.

| OF | 0.1 | 0.2 | 0.3 | 0.4 | 0.5 | 0.6 | 0.7 | 0.8 | 0.9 | 1.0 |
|----|-----|-----|-----|-----|-----|-----|-----|-----|-----|-----|
| sliding LPNet (mAP) | 75.2 | 75.8 | 76.1 | 76.3 | 76.5 | 76.6 | 76.9 | 77.2 | 77.5 | 77.6 |
| stable LPNet (mAP) | 75.4 | 75.9 | 76.3 | 76.4 | 76.8 | 77.1 | 77.3 | 77.6 | 77.6 | 77.7 |

Table 3 shows that, as the value of *OF* increases, the network noise is relatively suppressed, which leads to a certain degree of improvement in accuracy. When the *OF* of the gradient is given, the network accuracy also shows the result of the increasing gradient. When *OF* = 1, this situation is called the full observation state and the corresponding network accuracy is the highest.

*4.4. Dataset Rotation*

In Figure 2, we show the effect of the object at different rotation angles. The arrangement of feature maps corresponding to different rotation targets is the same, which effectively reduces the influence of rotation on the network accuracy. In training the encoded feature map, the network not only has the characteristics of the cartesian coordinates but also those of the log-polar coordinate, which enhances the robustness of the network.

In this section, an overall rotation transformation is carried out on the VOC2007 and VOC2012 datasets to test the anti-rotation performance of LPNet. For these two datasets, training and testing are carried out with rotations of 45°, 90°, and 180°. The original SSD network is also tested under the same experimental conditions.

Figure 7 shows the final experimental results. As the image in the dataset rotates by a certain angle, the accuracy of target detection also decreases. However, LPNet can reduce the impact of this rotation on accuracy. We choose to compare the accuracy of Figure 8a–d under the condition of *OF* = 1. As the image rotates, the mAP of the SSD300 (the blue straight line) decreases by about 13% at most. Compared to SSD300, the mAP of LPNet increases by up to 15.6%. Although rotation also reduces the mAP of LPNet, the mAP will not drop by more than 2% at most. By comparison of Figure 8e–h, we can also see that, compared to stable LP, sliding LP is more suitable for rotating object detection.

Figure 8 shows that, for a normal dataset (without rotation processing), the SSD network reaches an accuracy of 74.3%. However, at a certain angle rotation, the network adaptability decreases and the highest accuracy becomes 61.32%. The LPNet with log-polar transformation maintains the original accuracy without a decrease, and the network has strong robustness. For stable and sliding LPNet, under the normal dataset, the accuracy of stable LPNet is slightly higher than that of sliding LPNet. However, for rotating targets, sliding LPNet is more adaptable. Given that its center point is constantly moving, the network of the sliding LPNet is more suitable than the stable LPNet to capture the characteristics of various forms when the target rotates. We show some images as examples in Figure 9, which are rotated in several angles and detected by LPNet.

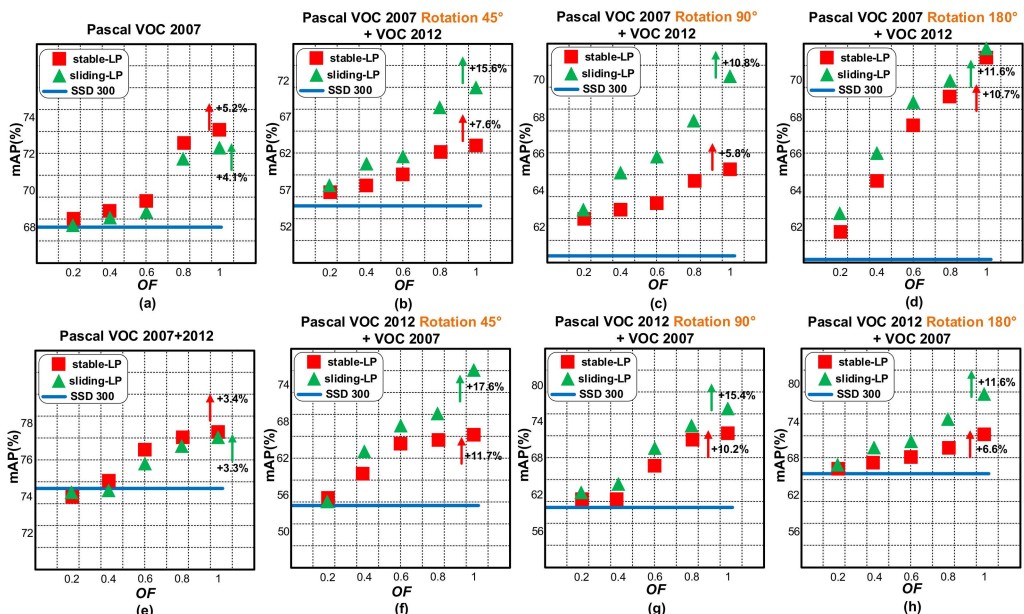

**Figure 8.** LPNet changes in network accuracy under different *OF*s for VOC2007, VOC2012, and rotation transformed datasets. Where, (**a**) and (**e**) represent experimental results in which LPNet is not added and the datasets are not rotated. (**b**) and (**f**) represent the experimental results of VOC2007 and VOC2012 respectively rotated by 45 degrees. (**c**) and (**g**) represent the experimental results of VOC2007 and VOC2012 respectively rotated by 90 degrees. (**d**) and (**h**) represent the experimental results of VOC2007 and VOC2012 respectively rotated by 180 degrees.

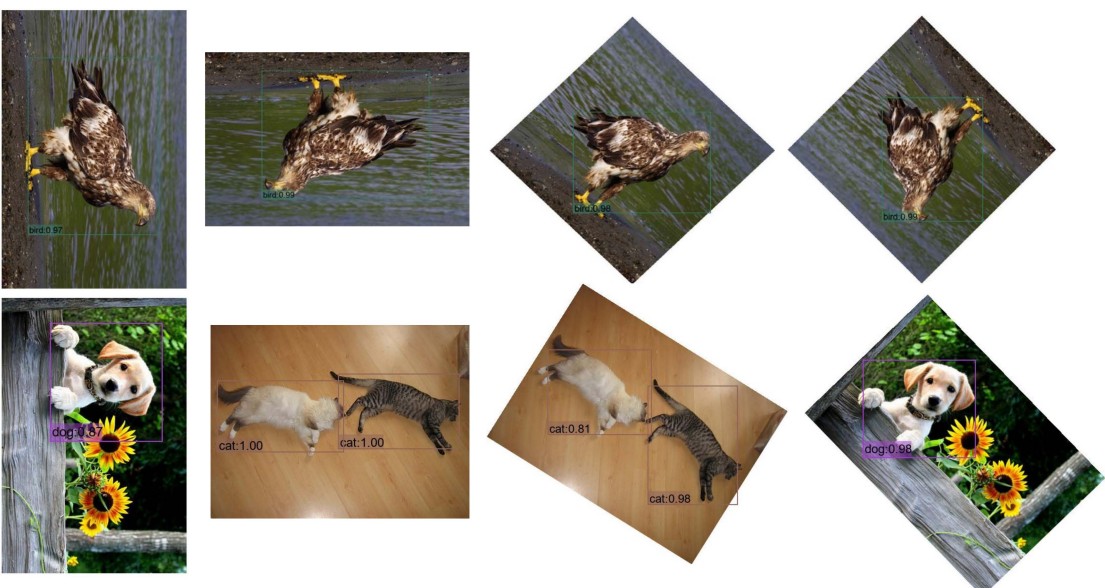

**Figure 9.** Images rotated in several angles detected by LPNet.

## 5. Conclusions and Future Works

In this study, we introduce log-polar transformation into the deep learning object detection and recognition network, which effectively reduces the influence of objects rotation on detection accuracy. However, further research is still necessary. This research selects relatively reasonable parameters but no experiments on the selection of step size $\lambda$, which must also be studied and discussed in future studies. We also did not conduct

training and testing on large datasets, such as the COCO. The network recall and other parameters also need testing in combination with different network frameworks.

Log-polar transformation is a method with a retina-like mechanism. This study uses LPNet to alleviate the influence of objects rotation and enhance the robustness of CNN. However, this mechanism is also used more widely in other aspects of object detection and recognition models in deep learning. The human eye has the advantages of variable resolution and scale transformation, which correspond to the attention mechanism of object detection and recognition network. This trend makes CNN more simple and human-like in the future.

**Author Contributions:** Conceptualization, J.C. and C.B.; methodology, C.B.; software, C.B.; validation, J.C., Y.C. and Q.H.; formal analysis, C.B.; investigation, C.C.; resources, J.C.; data curation, Y.C.; writing—original draft preparation, J.C. and C.B.; writing—review and editing, Q.H.; visualization, C.C.; supervision, Q.H.; project administration, C.B.; funding acquisition, J.C. All authors have read and agreed to the published version of the manuscript.

**Funding:** This work was supported in part by the funding of foundation enhancement program under Grant 2020-JCJQ-JJ-030, in part by the National Natural Science Foundation of China under Grant 61871031, Grant 61875012 and Grant 61905014.

**Data Availability Statement:** Not applicable.

**Conflicts of Interest:** The authors declare no conflict of interest.

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
