# Peer review of "LPNet: Retina Inspired Neural Network for Object Detection and Recognition"

_electronics, doi:10.3390/electronics10222883_

Round 1

Reviewer 1 Report

Line 118: Please elaborate SSD (this is the first time you are mentioning SSD, the abstract is not part of the main body).

Figure 2 is not properly described. For example, how do you get a 3D fully-connected layer (FC6 and FC7)?

Line 159: Equation numbers should be checked.

The proposed architecture is not clearly written. An interested reader cannot implement the architecture by reading the article. All the processes should be described. For example, the encoder and the decoder should be detailed.

Needs a brief description of the datasets.

Table 2: Are these accuracies (%)?

More critical analysis of the results is desired. 

Reviewer 2 Report

This paper deals with an exciting topic. The article has been read carefully, and some crucial issues have been highlighted in order to be considered by the author(s).

All the acronyms should be defined and explained first before using them such that they become evident for the readers.

The paper needs to be restructured in order to be precise.The Introduction and related work parts give valuable information for the readers as well as researchers. In addition recent papers should be added in the part of related work.

As it is real time application oriented, authors should care over the outcome of the proposed framework by meeting the future requirements too.

Representation of figures needs to be improved.

Grammatical errors should be validated. Most of the typos and incorrect grammars have been corrected, but it is still necessary to subject the paper to proofreading.

It would be good if security [1] and recognition [2] domains  would be reflected in future research or related work.

[1] Kwon, Hyun, et al. "Classification score approach for detecting adversarial example in deep neural network." Multimedia Tools and Applications 80.7 (2021): 10339-10360.

[2] Ko, Kyoungmin, et al. "SqueezeFace: Integrative Face Recognition Methods with LiDAR Sensors." Journal of Sensors 2021 (2021). 

Round 2

Reviewer 2 Report

This paper is worth for acceptance.